

# Reproductive pattern in the solanum mealybug, *Phenacoccus solani*: A new perspective

Jun Huang[1], Fuying Zhi[1,2], Juan Zhang[3], Muhammad Hafeez[1], Xiaowei Li[1], Jinming Zhang[1], Zhijun Zhang[1], Likun Wang[1] and Yaobin Lu[1]

[1] Institute of Plant Protection and Microbiology, Zhejiang Academy of Agricultural Sciences, Hangzhou, China
[2] College of Chemistry and Life Sciences, Zhejiang Normal University, Jinhua, China
[3] Institute of Garden Plants and Flowers, Zhejiang Academy of Agricultural Sciences, Hangzhou, China

## ABSTRACT

**Background**. The reproductive pattern of most scale insects is ovoviviparity. The solanum mealybug, *Phenacoccus solani* (Hemiptera: Pseudococcidae), is known as a thelytokous parthenogenetic species, but there is still debate about the reproductive strategies of this species.

**Methods**. Here, we investigated the oviposition characteristics of *P. solani* and used scanning/transmission electron microscopy and RNA-seq to identify the differences between two types of eggs.

**Results**. We found that *P. solani* laid two types of eggs in one batch, with no significant difference in apparent size: one with eyespots that hatched and another without eyespots that failed to hatch. Furthermore, the physiological and molecular differences between the two types of eggs were highly significant. KEGG enrichment analysis revealed significant enrichment for the JAK-STAT, Notch, Hippo, and Wnt signaling pathways and dorsoventral axis formation, wax biosynthesis, cell cycle, insulin secretion, and nitrogen metabolism pathways. The results suggest that the embryo of the egg undergoes development inside the mother and only a short molting period outside the mother.

**Discussion**. Ovoviviparous species produce eggs and keep them inside the mother's body until they are ready to hatch, and the offspring exits the egg shell during or immediately following oviposition. Therefore, we suggest that the reproductive pattern of *P. solani* can be described as ovoviviparity.

# INTRODUCTION

The reproductive strategies of most insects have been described with three main patterns: oviparity, viviparity and ovoviviparity (*Meier, Kotrba & Ferrar, 1999*; *Wheeler, 2003*; *Gullan & Granston, 2014*). In oviparous species, egg development occurs in the external environment after oviposition, and hatching occurs outside the mother's body, whereas in viviparous species (which are relatively rare among insects), egg development occurs inside the mother's body, which provides gas exchange and, more importantly, nourishment for the embryos which are born alive (*Andrews & Rose, 1994*; *Tworzydło et al., 2013*). Based on nutritional relationships between maternal and embryonic organisms, two

Corresponding authors
Jun Huang,
junhuang1981@aliyun.com
Yaobin Lu, luybcn@163.com

modes of viviparity are recognized: matrotrophic and lecidotrophic (*Ostrovsky et al., 2016*). Ovoviviparity is in fact a specific type of viviparity where developing eggs are retained within the body of the mother, and the offspring are nourished by the reserve materials accumulated in the eggs during oogenesis (*Blackburn, 1999*; *Gaino & Rebora, 2005*; *Lodé, 2012*). Therefore, the recent view is that there are truly only two main reproductive strategies: viviparity and oviparity (*Ostrovsky, 2013*; *Ostrovsky et al., 2016*). In previous studies, some insects, such as cockroaches (*Warnecke & Hintze-Podufal, 1996*), aphids (*Ortiz Rivas, Moya & Torres, 2004*), tsetse flies (*Meier, Kotrba & Ferrar, 1999*), thrips (*Kranz et al., 2002*), and scale insects (*Gavrilov & Kuznetsova, 2007*; *Ngernsiri et al., 2015*), were described as Ovoviviparous species. Scale insects (Hemiptera: Sternorrhyncha: Coccoidea), like many Hemiptera, feed on sap drawn directly from the plant vascular system and secrete a waxy coating for defense; in addition, many scale insect species are major quarantine pests of agricultural or ornamental plants in tropical/subtropical climates as well as in greenhouses in temperate zones worldwide (*Gullan & Kosztarab, 1997*; *Gavrilov-Zimin, 2018*). All previous relevant studies suggest that the phenomenon of ovoviviparity is widely distributed among scale insects (*Tremblay, 1997*; *Gavrilov & Kuznetsova, 2007*). *Gavrilov-Zimin (2018)* provided an overview of the distribution of different variants of ovoviviparity/viviparity among scale insect families and demonstrated that the evolution of scale insects shows multiple cyclic conversions of the oviparous reproduction pattern to ovoviviparity/viviparity, with the appearance of new and interesting adaptations for egg protection. In other words, the reproductive modes of scale insects may be rich and variable; however, the understanding of the course of evolution of reproductive patterns in these insects is not complete (*Gavrilov & Kuznetsova, 2007*). Therefore, the identification of reproductive patterns has great heuristic value in terms of both reproductive and evolutionary biology.

The solanum mealybug, *Phenacoccus solani* (Hemiptera: Pseudococcidae), is native to North America (*Chatzidimitriou et al., 2016*) and is a newly recorded species in China. Furthermore, this mealybug species is quite polyphagous and considered to be a major threat to agricultural production and the ecological environment, causing significant problems (*Ben-Dov, 2005a.*; *Zhi et al., 2018*). *P. solani* is a thelytokous parthenogenetic species, and no male individuals are found in its populations (*Lloyd, 1952*; *Ben-Dov, 2005b*; *Zhi et al., 2018*). Regarding the birth strategies of *P. solani*, McKenzie reported that this species was viviparous (*McKenzie, 1967*); however, *Kosztarab (1996)* and *Chatzidimitriou et al. (2016)* considered this species to be ovoviviparous. *Vennila et al. (2010)* found that in another species of mealybug in the same genus, parthenogenesis via ovoviviparity (96.5%) was dominant over oviparity (3.5%). Many scholars have found that the hatching period of eggs laid by mealybugs is very short and that the hatching process is relatively concealed (beneath the abdomen), as a result, short and concealed hatching has been suggested as the main reason for the divergence of reproductive modes in mealybugs (*Tremblay, 1997*; *Lagowska & Golan, 2009*; *Vennila et al., 2010*; *Zhi et al., 2018*). According to our previous observations, we believe that the reproductive mode in mealybugs, at least in *P. solani,* is complex and not simple to define.
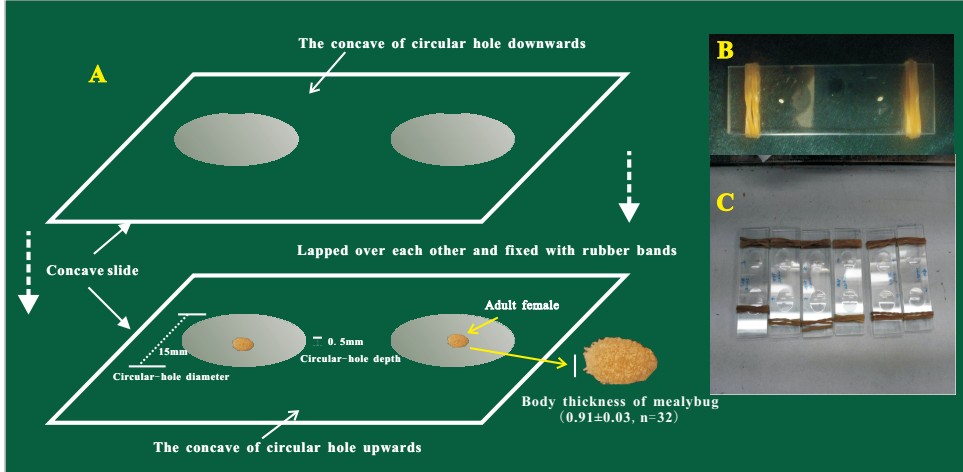

**Figure 1   The device used for the observation of egg-laying and egg hatching of adult females of** *Phenacoccus solani* **(Double-concavity slide method).** (A) Position and operation diagram. Two concave slides overlapped seamlessly, and adult female was placed into the middle of the hole of the concave slide; concave slides were fixed with rubber bands at both ends. (B) The final observation device contained two separate adult females. (C) Multiple devices together.

Here, we investigated the oviposition characteristics of *P. solani* and used scanning electron microscopy (SEM) and transmission electron microscopy (TEM) to distinguish between two types of eggs (i.e., reproductive products putatively considered to be eggs) laid in one batch. Finally, we further investigated miRNA and mRNA expression in the two types of eggs. We aimed to answer the following questions: (1) What life stage is *P. solani* laying–an egg, a nymph, or something in between? (2) If different reproductive "products" coexist, do they differ in appearance, physiology and molecular biology? (3) According to the above findings, what is the reproductive mode of *P. solani*?

## MATERIALS & METHODS

### Oviposition characteristics

The oviposition process of female adults of *P. solani* was observed on two transparent, single-well, concave slides (Fig. 1) (length * width = 76.2 mm * 25.4 mm; the thickness of each concave slide was 1.3 mm; the diameter and depth of the circular hole were 15 mm and 0.5 mm, respectively). The steps of the procedure were as follows: (1) a concave slide was placed face up on a flat table and a female adult was gently placed into the middle of the hole of the concave slide with a brush; (2) quickly another concave slide was placed face down, on top of the bottom concave slide; and (3) finally, rubber bands were used to bind the ends of the two concave slides to secure them. Preliminary experiments showed that the thickness of female adults was 0.91 ± 0.03 mm, so the above observation method did not harm the bodies of female adults. The egg hatching process was further tracked and observed under a NikonSMZ1500 zoom stereomicroscope (Nikon, Japan), and photographs were taken every 5 min from the start of the egg laying process.

## Egg morphology

The number, length, width and hatching rate of eggs with eyespots (hereafter eggs with-ES) or eggs without eyespots (hereafter eggs without-ES) were further observed in the same batch of eggs under a NikonSMZ1500 zoom stereomicroscope. There were two kinds of treatments: one employed the concave slide method (placing a female adult between two concave slides as mentioned above), and the other employed the blade method, i.e., placing a female adult on detached potato leaves, with the petiole wrapped with defatted cotton to maintain leaf freshness and the placing the whole treated leaves in Petri dishes (diameter = 9.0 cm, thickness = 1.4 cm). The oviposition of female adults was observed every 30 min from 9.00 a.m. to 4.00 p.m, and the numbers of the two kinds of eggs were counted. Each female adult was biometrically tested once, and each treatment was repeated 15 times. A total of 25 eggs were randomly selected from the two kinds of eggs, and their lengths and widths were measured. After 72 h, the hatching of the two kinds of eggs was observed. Eggs with-ES hatched 114 individuals, while eggs without-ES hatched 45 individuals. The experiment was carried out in an artificial climate chamber with a temperature of $27 \pm 1 \,^{\circ}\text{C}$, a humidity of $70\% \pm 5\%$ and a photoperiod of 16 L: 8D.

## Does the mother's body affect the hatching of eggs?

Two treatments were established: (1) eggs with-ES were incubated under the mother's body (Fig. 2A), and (2) eggs with-ES were artificially removed from the mother and kept on the concave slide (Fig. 2C). Then, the female adults were continuously observed through a NikonSMZ1500 zoom stereomicroscope every five minutes. Continuous stretching of the abdomen by female adults indicated that they were about to lay eggs. For treatment 2, the upper concave slide was removed immediately, and the mother was carefully removed with an insect pin. Hatching time was recorded as the time from when the female adult laid eggs to the time when the eggshell was completely detached. Each egg was bioassayed once; experiment 1 included 30 replicates, and experiment 2 included 47 replicates.

## Microscopic differences between the two kinds of eggs
### Egg surface and internal structure

The surface and internal structure of the two kinds of eggs were observed by SEM (SU8010, Hitachi, Japan) and TEM (H7650, Hitachi, Japan). The collected eggs with and without ES were pretreated with liquid nitrogen immediately. The follow-up procedures followed the instrument operation methods for the scanning electron microscope. TEM was performed as follows: (1) fixation, eggs were immersed in 2.5% glutaraldehyde fixative solution and then rinsed with buffer solution; (2) gradient dehydration, the fixed samples were dehydrated with an ethanol; (3) gradient osmosis, the samples were permeated with a mixture of acetone and Spurr resin penetrant (1:1); (4) embedding and polymerization, 100% Spurr embedding agent was added, followed by polymerization for 24 h; (5) ultrathin sectioning, the samples were cut to approximately 90 nm; and (6) observation and photography.

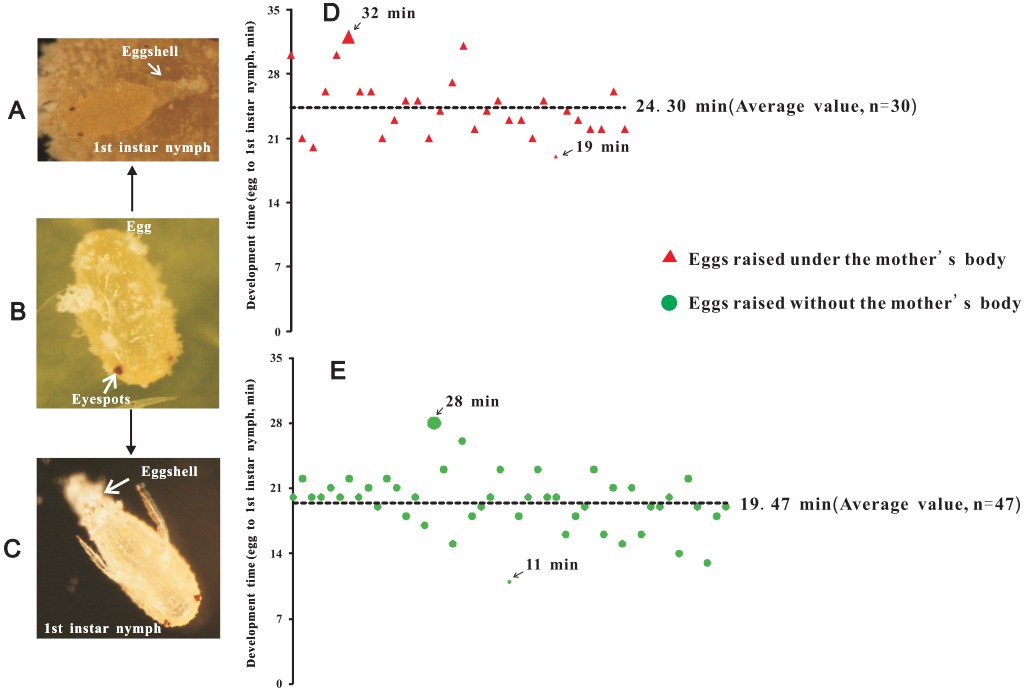

**Figure 2** **The hatching time of eggs beneath the abdomen of adult female or in isolation.** (A) Egg hatching beneath the abdomen of adult female, at the point at which the eggshell detached; (B) Eggs with eyespots; (C) Egg hatching in isolation (without mother's body); (D) Hatching time of eggs beneath the abdomen of adult female; (E) Hatching time of eggs without the mother's body or in isolation.

### RNA sequencing and data analysis

*RNA extraction, library construction and RNA sequencing.* Samples of newly laid eggs (with-ES or without-ES) were collected and immediately placed in a 0.5 MlEP tube and then frozen in liquid nitrogen for RNA extraction. Total RNA was extracted from 6 samples, and a library was constructed as previously described (*Yin et al., 2018*). The libraries were sequenced on the Illumina HiSeq X Ten platform, and 150 bp paired-end reads were generated. Raw data (raw reads) in fastq format were first processed using Trimmomatic (*Bolger, Lohse & Usadel, 2014*). Reads containing poly-N stretches and reads oflow quality were removed to obtain clean reads. After removing adaptor and low-quality sequences, the clean reads were assembled into expressed sequence tag clusters (contigs) and de novo assembled into transcripts by Trinity (*Grabherr et al., 2011*) (version: 2.4) with the paired-end method. The longest transcript was chosen as a unigene based on similarity and length for subsequent analyses. Raw data were deposited in the National Center for Biotechnology Information (NCBI) Sequence Read Archive (SRA) (https://www.ncbi.nlm.nih.gov/sra) under accession number PRJNA554708.

### Unigene de novo assembly, functional annotation and data analysis

Transcriptome sequencing and analysis were conducted by OE Biotech Co., Ltd. (Shanghai, China). The functions of the unigenes were annotated by alignment of the unigenes with the NCBI nonredundant (NR), SwissProt, EuKaryotic Orthologous Groups (KOG), Gene

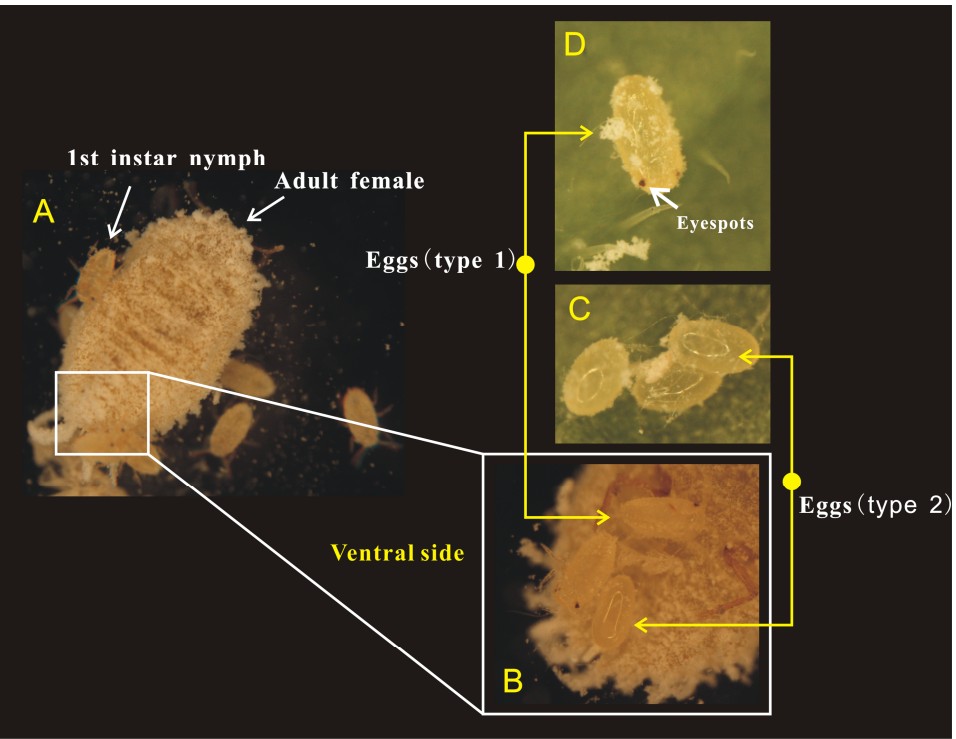

**Figure 3** **Adult female of *Phenacoccus solani*. laid two types of eggs in one batch, one with eyespots (type 1), and another without eyespots (type 2).** (A) Adult female laying eggs, and some eggs have rapidly hatched into 1st instar nymphs; (B) Newly laid two types of eggs below the abdomen of adult female; (C) The eyespots of eggs (type 2) were not visible; (D) Eggs (type 1) before hatching, the eyespots were clearly visible.

Ontology (GO) and Kyoto-Encyclopedia of Genes and Genomes (KEGG) databases. Differentially expressed unigenes (DEGs) were identified using the DESeq (*Anders & Huber, 2013*) functions "estimate size factors" and "nbinom test". A *p* value <0.05 and fold change >2 or <0.5 were set as the thresholds for significant differential expression. Hierarchical cluster analysis of DEGs was performed to explore transcript expression patterns, and KEGG pathway enrichment analysis of DEGs was performed using R based on a hypergeometric distribution.

## RESULTS

### Oviposition characteristics

Female adults of *P. solani* produced eggs in one generation through thelytokous parthenogenesis, and the eggs were long and oval, similar to those described by *Zhi et al. (2018)*. Female adults secreted silken wax, but it never formed an ovisac. After egg laying, the eggshell was detached from the nymph, followed by the appearance of 1st-instar nymphs that quickly crawled away from the lower part of the mother. Moreover, in a batch of eggs laid by female adults of *P. solani*, two kinds of eggs with distinct morphological differences appeared (Figs. 3A/3B): two reddish-brown eyespots were seen on one type

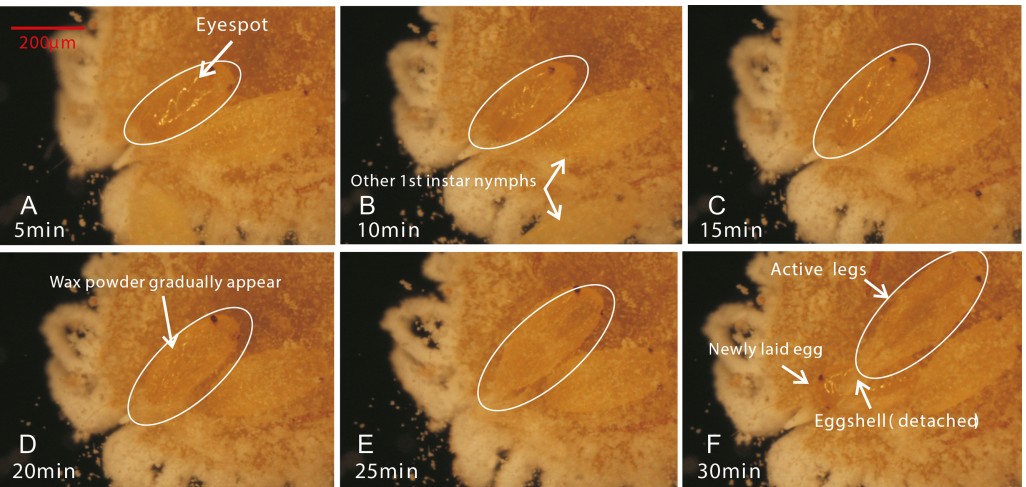

**Figure 4** **The hatching process of eggs.** (A) After the first 5 minutes, eggs began to show considerable peristalsis beneath the mother; (B/C) At 10–15 minutes, the eggshell was gradually detached and wax powder appeared on the surface of body; (D) At 20 min, antennae and feet were starting to become visible; (E/F) The hatching was basically completed, while the detached eggshell could be seen at the abdominal end of the 1st instar nymph.

of egg (eggs with-ES; Fig. 3D), while the other type did not display these eyespots (eggs without-ES; Fig. 3C). The hatching process of eggs with-ES was as follows: (1) in the first 5 min, the eggs began to show considerable peristalsis beneath the mother, and at 10–15 min, the eggshell was gradually detached, and wax powder appeared on the surface of the body; and (2) at approximately 20 min, antennae and feet began to appear, and the hatching process was generally completed within 25 min, while the detached eggshell could be seen at the abdominal end of the 1st instar nymph (Fig. 4).

## Egg morphology

In the present study, we observed that female adults could lay two types of eggs, those with-ES and those without-ES, in either treatment. The ratio of eggs with-ES was significantly larger than that of eggs without-ES (on concave slides, $91.56 \pm 2.14$ vs. $8.44 \pm 2.14$, respectively, $\chi 2 = 86.01$, $n = 15$, $p < 0.001$; on leaves, $87.31 \pm 2.90$ vs. $12.69 \pm 2.90$, respectively, $\chi 2 = 13.07$, $n = 15$, $p < 0.001$) (Fig. 5A). The eggs were long and elliptical, with lengths of $0.320 \pm 0.006$ mm (with-ES) and $0.305 \pm 0.008$ mm (without-ES) ($t = 1.42$, $n = 25$; Fig. 5C) and widths of $0.146 \pm 0.002$ mm (with-ES) and $0.147 \pm 0.003$ mm (without-ES) ($t = 0.24$, $n = 25$; Fig. 5D). Furthermore, the hatching rate of eggs with-ES was 100%, and no eggs without-ES hatched (Fig. 5B).

## Does the mother's body affect the hatching of eggs?

To investigate the extent to which the mother's body affected the hatching of eggs, eggs without-ES were removed from the area beneath the mother's body after they were laid. Therefore, we further tested whether eggs with-ES hatched after they were removed from the area beneath the mother's body. We found that eggs with-ES could hatch normally under any treatment (Figs. 2A–2C), but the hatching times were different. Inside the

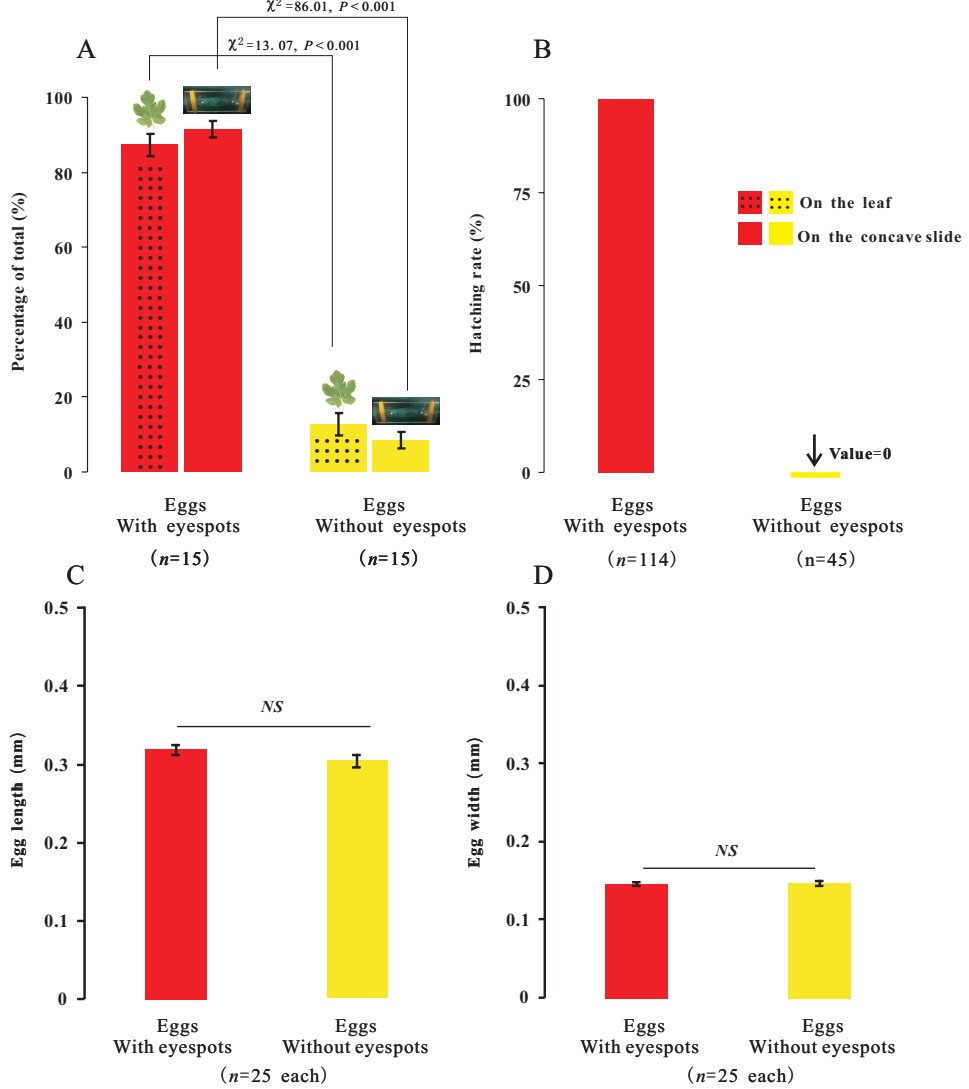

**Figure 5 The percentage, morphology and hatching of the two types of eggs with different treatments.** Percentage of the total for the two types of eggs with the treatment of placing female adults on leaves or concave slide (A), and hatching rate of eggs (B); the difference between the two types of eggs with same treatment was analyzed using Chi-SquareTest. The length (C) and width (D) of two types of eggs; the difference between the two types of eggs was analyzed using T-Test, and "*NS*" on the two bars indicate not significantly different from each other ($p > 0.05$).

mother's body, the hatching time was $24.30 \pm 0.60$ min (Fig. 2D), but when the mother's body was removed, the hatching time was reduced by nearly 5 min to $19.47 \pm 0.45$ min (Fig. 2E). Therefore, we suggest that the mother's body has no effect on the success of egg hatching, or it could be inferred that the failure of eggs without-ES to hatch was largely due to internal factors. Many species of scale insects secrete abundant wax and form an ovisac that covers eggs and prevents their adhesion, but in some species, ovisacs are not built, and the time of egg development outside of the maternal body is decreased
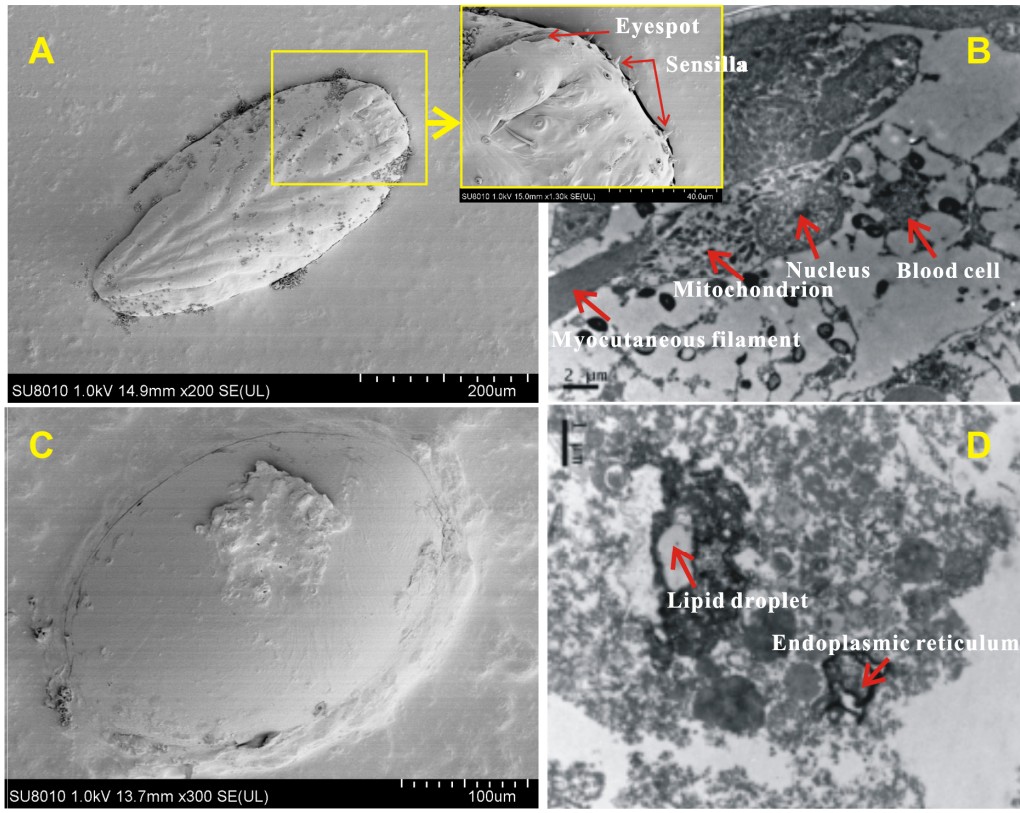

**Figure 6** **The surface and internal structure of eggs with eyespots and without eyespots.** The surface of eggs with eyespots (A) and without eyespots (C) was observed by scanning electron microscope, and the internal structure of eggs with eyespots (B) and without eyespots (D) was observed by transmission electron microscope.

(*Gavrilov-Zimin, 2018*). *P. solani* belongs to the latter group, secreting a small amount of wax and never forming an ovisac. Thus, it is not surprising that the eggs of *P. solani* hatched so quickly, especially when the eggs were isolated.

## Microscopic differences between the two types of eggs
### Egg surface and internal structure
The surface and internal structure of eggs with and without-ES were observed using SEM and TEM. The contour of appendages could be clearly seen across the eggshell, and the bristles, tubular glands and six conical receptors (which were symmetric, with three on each side) could be seen on the surfaces of eggs with-ES (Fig. 6A); moreover, complete blood cells, cytoplasm, mitochondria and myocutaneous filaments were observed inside (Fig. 6B). Eggs without-ES had a smooth surface (Fig. 6C) and contained only lipid droplets, endoplasmic reticulum and free ribosomes (Fig. 6D).

### RNA sequencing and data analysis
Illumina sequencing generated approximately 45 M reads per sample after the removal of low-quality reads. These reads were assembled randomly and produced 55,558 unigenes

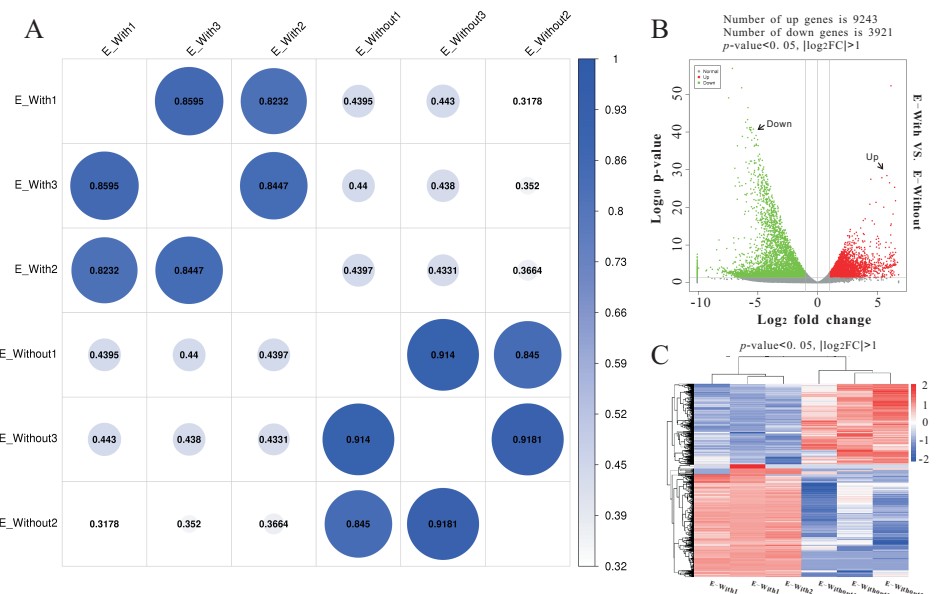

**Figure 7 RNA-seq to distinguish the differences between two types of eggs.** Heat-map coefficient matrix (A); the abscissa indicates the sample name, the ordinate indicates the corresponding sample name, and the color indicates the correlation coefficient. The closer the correlation coefficient is to 1, the higher the similarity in expression patterns between samples. Volcano plots of different differentially expressed unigenes (DEGs) between two groups. Green dots indicate down-regulated unigenes, red dots indicate upregulated unigenes, and grey dots indicate no differential unigenes (B). Cluster analysis of DEG levels. Expression differences are shown in different colors. Red and blue indicate up-regulation and down-regulation, respectively (C).

with an N50 of 1,026 nt. After annotating unigenes with several databases and calculating the expression of unigenes as fragments per kilobase of exon model per million reads mapped (FPKM), correlation coefficients between samples were calculated and used to estimate biological repeatability and differences between groups. The correlation coefficient of 3 biological replicates in the group with-ES and the group without-ES was >0.8, and the sample correlation coefficient between these two groups was only 0.4, showing an obvious difference between these two groups (Fig. 7A). DEGs were identified and screened with the thresholds of $p < 0.05$ and fold change > 2 (or fold change < 0.5) (Figs. 7B/7C). There were 13,164 DEGs between the with-ES and without-ES groups, including 9,243 up regulated DEGs and 3,921 down regulated DEGs. The DEGs are shown with a volcano plot, and a heat map was generated from hierarchical cluster analysis to show the expression patterns of the DEGs (Fig. 7B). KEGG enrichment analysis of the DEGs was performed to determine the main pathways associated with these DEGs. We found that the JAK-STAT, Notch, Hippo, and Wnt signaling pathways and dorso-ventral axis formation, wax biosynthesis, progesterone-mediated oocyte maturation, cell cycle, eukaryote ribosome biogenesis, insulin secretion, and nitrogen metabolism pathways were significantly enriched (Fig. 8).

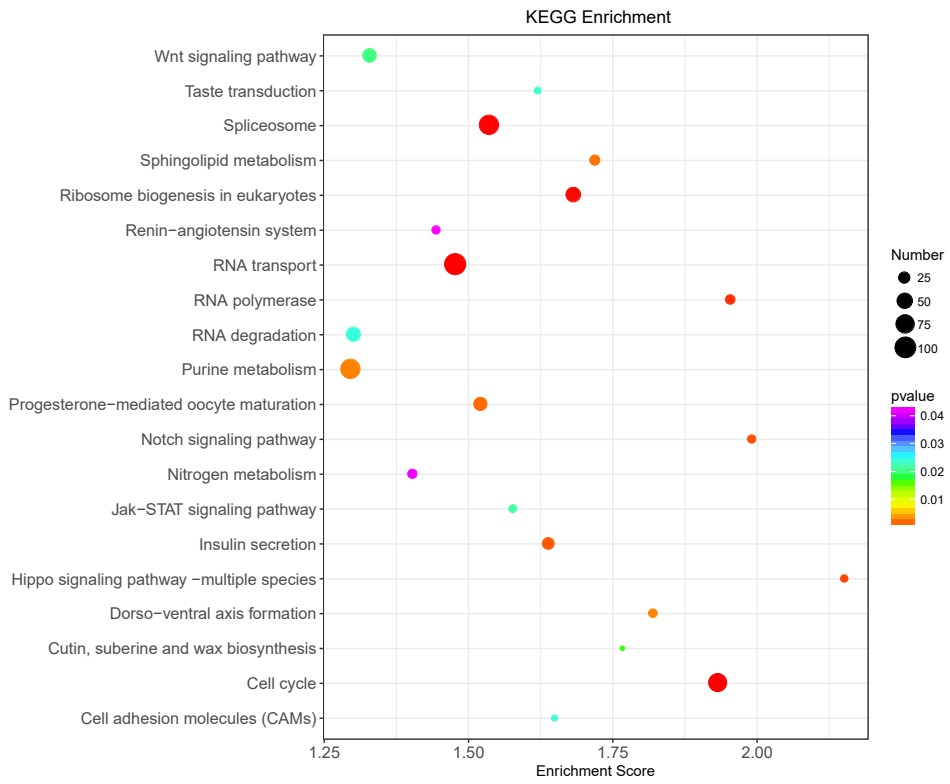

**Figure 8  KEGG pathway enrichment analysis of differentially expressed genes (DEGs).** Only the top 20 pathways in KEGG enrichment function were listed in the figure. The ordinate is the name of the KEGG metabolic pathway, and the abscissa is the Enrichment Score to the pathway. The larger the bubble is, the more different the number of Unigene is contained, and the bubble color changes from purple-blue-green-red, with smaller Enrichment $p$ value and greater significance.

## DISCUSSION

### Oviposition and hatching characteristics

In the present study, we found eggs laid by parthenogenetic *P. solani*, but soon emerged 1st-instar nymphs following hatch. Because this hatching process is fast (<30 min), researchers might think that 1st-instar nymphs are produced by *P. solani* female adults. A similar phenomenon was reported in another invasive mealybug, *Phenacoccus solenopsis*, and because both eggs and 1st-instar nymphs were found in oocysts, *Vennila et al. (2010)* considered the scale insect to be able to reproduce in both ways, i.e., via ovoviviparity and oviparity. In *Coccus hesperidum*, naked nymphs appeared from the vulvar orifice, but thin eggshells were shown to remain in the female reproductive tract (*Hagan, 1951*), and in *P. solani*, the eggshells were kept beneath the mother.

In *Matsucoccus matsumurae*, there are two types of eggs, those with and without eyespots, but the eggs with eyespots are similar to those without eyespots, and the egg types show different developmental periods with normal hatching (*Xie et al., 2014*). However, *P. solani* laid two types of eggs in one batch, with no significant difference in apparent size: one with eyespots that hatched and another without eyespots that failed to hatch. Generally, eggs

are under stress from external factors, which may prevent them from hatching properly. For example, some of the eggs laid by heat-treated females of *Nilaparvata lugens* were unable to hatch due to failure during blastokinesis (*Lee & Roger, 1987*). Further research revealed that yeast-like symbiotes in *N. lugens* play an important role in the embryonic and postembryonic development of eggs, especially the ventral differentiation of the embryo (*Wilkinson & Ishikawa, 2001*; *Nan et al., 2016*). The symbiotic bacteria in the bodies of most mealybug subfamily insects are *Tremblaya princeps* (*Gruwell et al., 2010*), play a substantive role in the host plant specificity of their hosts (*Baumann, 2005*; *Hansen & Moran, 2014*), and are correlated with host development (*Huang, Zhao & Lu, 2015*); therefore, we suggest that the absence of symbiotes might explain the presence of nondeveloping eggs. Another type of insect egg that does not hatch is the nutritive egg common in social insects, such as ants. The nutritive eggs of ants are unfertilized eggs that cannot hatch and are eaten in colonies containing a queen (*Heinze, Trunzer & Oliveira, 1996*; *Heinze et al., 1999*). However, for this parthenogenetic and thelyotokous species of mealybug, the factors causing *P. solani* to lay eggs that cannot hatch require further study.

## Physiological and molecular differences between the two types of eggs

According to these observations of egg surface and internal structure, we suggest that eggs with-ES are alive, with features such as conical receptors, blood cells and myocutaneous filaments, and that they still have an eggshell and are close to hatching, even though they no longer resemble an egg on a microscopic level. Although some important organelles such as endoplasmic reticulum and free ribosomes were present in the eggs without-ES, mitochondria had never been found. Mitochondria are a special organelle that contain their own genomes and plays an important role in oocyte maturation and embryo development (*Ferenz, 1993*; *Lieber et al., 2019*). Moreover, mitochondria are inherited only in the maternal line, i.e., mitrochondrial DNA is passed only through the mother's egg cell (*Ma, Xu & O'Farrell, 2014*; *Lieber et al., 2019*). Lack of mitochondrial redistribution in cytoplasm was a sign of immature oocyte and was closely related to low developmental of eggs (*Bavister & Squirrell, 2000*). If mitochondria were missing, it could be fatal to the development of egg cell and later embryonic development. Therefore, we hypothesized that the development of egg cell might be arrested for the eggs without-ES in the absence of mitochondria.

We further determined the differences between eggs with and without ES at the molecular level. The result revealed that the differences in terms of unigene expression between two types of eggs were highly significant, and we also found that the JAK-STAT, Notch, Hippo, and Wnt signaling pathways and some important pathways related to metabolism and nutrition were significantly enriched. Recently, the JAK-STAT, Notch, Hippo and Wnt signaling pathways were found to independently or interactively participate in the regulation of egg production (*Hombria & Brown, 2002*; *McGregor, Xi & Harrison, 2002*). For example, mutual antagonism between the Notch and JAK/STAT signaling pathways provides a crucial facet of follicle cell patterning and ultimately helps establish the polarity of the egg chamber (*Assa-Kunik et al., 2007*), and the Hippo pathway controls polar cell

specification by repressing Notch activity (*Chen et al., 2011*). Moreover, some of these important signaling pathways are involved in aspects of cell development and metabolic function, such as dorsal-ventral axis formation, wax biosynthesis, insulin secretion and nitrogen metabolism.

## CONCLUSIONS

Although the reproductive mode of *P. solani* has been described previously, there is still no clear agreement on its definition. We found no differences between the two types of eggs by visual observation, but the physiological and molecular differences were highly significant. The results suggest that the embryonic development of eggs with-ES is complete when the eggs are laid beneath the abdomen, i.e., the embryo of the egg develops inside the mother. However, the embryonic development of eggs without-ES seems to be incomplete. Although there are no direct data on the entire process of embryonic development, we can at least be sure that the cell development and physiological metabolism of eggs without-ES are hindered or arrested. Moreover, we found that eggs with-ES begin to hatch and shed their eggshell (immediately) after leaving the mother's body: i.e., this species lays eggs and does not experience live birth. Ovoviviparous species oviposit eggs at an advanced stage of embryological development, and the larva exits the eggshell during or immediately following oviposition (*Meier, Kotrba & Ferrar, 1999*). Therefore, we suggest that the reproductive pattern of *P. solani* can be described as ovoviviparity.

## ACKNOWLEDGEMENTS

We would like to thank He Ziyi (Senior Experimentalist, Nanjing Agricultural University), and Liao Zhenfeng (Zhejiang Academy of Agricultural Sciences) for their support of electron microscope samples making and photos taking, and Dr. Wenya Lu and Dr. Fan Zhang (OE Biotech, Shanghai, China) for assistance with the bioinformatics analysis of the sequencing data. We are so grateful to Zhou Zhongshi (Chinese Academy of Agricultural Sciences), Jiang Mingxing (Zhejiang University), Xu Yijuan (South China Agricultural University), Ma Jun (Guangzhou Customs), and Fu Jianwei (Fujian Academy of Agricultural Sciences) for their comments and suggestions on the manuscript.

### Funding

This work was supported by the National Natural Science Foundation of China (Nos 31772234, 31801801) and Basic Public Welfare Research Project of Zhejiang Province (No. LGN20C140004). The funders had no role in study design, data collection and analysis, decision to publish, or preparation of the manuscript.

### Grant Disclosures

The following grant information was disclosed by the authors:
National Natural Science Foundation of China: 31772234, 31801801.
Basic Public Welfare Research Project of Zhejiang Province: LGN20C140004.

## Competing Interests

The authors declare there are no competing interests.

## Author Contributions

- Jun Huang conceived and designed the experiments, performed the experiments, analyzed the data, prepared figures and/or tables, authored or reviewed drafts of the paper, and approved the final draft.
- Fuying Zhi performed the experiments, analyzed the data, prepared figures and/or tables, authored or reviewed drafts of the paper, and approved the final draft.
- Juan Zhang performed the experiments, prepared figures and/or tables, authored or reviewed drafts of the paper, and approved the final draft.
- Muhammad Hafeez, Xiaowei Li, Jinming Zhang, Zhijun Zhang and Likun Wang analyzed the data, authored or reviewed drafts of the paper, and approved the final draft.
- Yaobin Lu conceived and designed the experiments, authored or reviewed drafts of the paper, and approved the final draft.

## DNA Deposition

The following information was supplied regarding the deposition of DNA sequences:

The Illumina HiSeq X Ten was used to sequence the libraries and generated 150bp paired-end reads. Raw data are available at the National Center for Biotechnology Information (NCBI) in the Sequence Read Archive (SRA): PRJNA554708.

## Data Availability

The raw measurements are available in the Supplementary Files.

## Supplemental Information

Supplemental information for this article can be found online at http://dx.doi.org/10.7717/peerj.9734#supplemental-information.

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
