# Peer review of "Reproductive pattern in the solanum mealybug, Phenacoccus solani: A new perspective"

_PeerJ, doi:10.7717/peerj.9734_

## Round 0.1 · original submission · Major Revisions

Dear Dr. Huang and colleagues:

Thanks for submitting your manuscript to PeerJ. I have now received three independent reviews of your work, and as you will see, the reviewers raised some concerns about the research. Despite this, these reviewers are optimistic about your work and the potential impact it will lend to research studying reproductive biology of the solanum mealybug. Thus, I encourage you to revise your manuscript, accordingly, taking into account all of the concerns raised by the reviewers.

Please address the issue raised over viviparity and oviparity. Also, ensure that other proper terminology is used throughout the manuscript.

There are many minor problems pointed out by the reviewers, and you will need to address all of these and expect a thorough review of your revised manuscript by these same reviewers.

I agree with the concerns of the reviewers, and thus feel that their suggestions should be adequately addressed before moving forward.

Therefore, I am recommending that you revise your manuscript, accordingly, taking into account all of the issues raised by the reviewers.

I look forward to seeing your revision, and thanks again for submitting your work to PeerJ.

Good luck with your revision,

-joe

Reviewer 1 ·

Basic reporting

The paper by Huandg el al. describes the reproductive modes of the solanum mealybug. I found the paper very interesting and adding new important data to our knowledge about developmental and evolutionary biology of insects. Solanum mealybug is a relatively new species in China and can cause important damaged to the crops.
The paper is written well, generally in proper English and is accompanied by nice figures that nicely complement the text.

There is one main point of the criticism, which I want Authors to address:
According to the recent views, there are TWO main reproductive strategies: viviparity and ovoparity. Ovoviviparity is in fact a specific type of the viviparity where developing eggs are retained within the body of the mother. Based on nutritional relationships between maternal and embryonic organisms two modes of viviparity are recognized: matrotrophic and lecidotrophic one. Please check the recent literature by Blackburn (1999) and Ostrovsky 2013 and Ostrovsky et al. 2016 on the proper terminology. Also the recent papers by Bilinski and his colleagues (Bilinski et al., 2017, 2018, 2019, Jaglarz et al., 2018, 2019; Tworzydlo et al., 2013, 2019) describing various aspects of viviparity in earwigs could be beneficial. Please correct and rewrite the first paragraph of the Introduction.
In line 75 Authors write that viviparity has been reported in Arixeniaesau and cite the paper by Benoit t al (2015). First of all, there should be Arixenia esau, secondly, the paper of Benoid focuses on the tsetse fly, and most importantly there are much more numerous examples of viviparous insects, even within earwigs. Please check carefully and correct that fragment.

Line 78 – please correct the sentence: “(…) young are produced by means of eggs (…)” is quite confusing. It should rather read: “The offspring is nourished by the reserve materials accumulated in the eggs during oogenesis (…).

Line 86: change “scale species” into “”scale insect species”.
I also think that a comprehensive English polishing and language editing will improve the manuscript.

Experimental design

The data presented in the paper meet all standards.
The Methods are presented properly with all sufficient details.

Validity of the findings

I found the findings interesting and well presented.

Additional comments

Taking into account all the aforementioned data, I strongly recommend acceptance of the paper after some minor corrections.

·

Basic reporting

The article is overall well-written, but there are several areas where the language needs to be clarified or corrected. These are especially concentrated in the figures and figure legends. Specific suggestions are listed below.

The authors have listed several of the key citations relevant to the field, including Blackburn 1999 and the publications by Gavrilov-Zimin. However, as described below, their findings are not consistent with the definitions of ovoviviparity provided in these references. In addition, as suggested in the language changes below, it is not clear which of the previous publications they have cited have already studied the same species, P. solani.

The authors have done a very nice job of providing the raw data, videos, and sequence information that will make this publication useful and relevant for future research. One note: the authors describe the process of assembling a de novo transcriptome for this species. They provide the raw data via SRA, but the de novo transcriptome should also made available. This is a substantial contribution the authors provide, and should be described more fully, unless this specific P. solani assembly has already been described elsewhere, in which case that needs to be referenced.

Suggested language changes:

- the word egg should be clarified throughout the manuscript. In insects, the word egg can refer to the structure that starts as a developing oocyte within the mothers body, as well as the egg stage of an insect lifecycle. The authors use the term inconsistently throughout the manuscript, for example, they state that in viviparous species "egg development occurs inside the mother's body" (line 72), but that in ovoviviparous insects "young are produced by means of eggs, not the body of the mother" (line 78). In order to clarify their meaning, they should restate the latter to say "offspring exit the mother's body during the egg stage, rather than during the larval stage".

- change "ovoviviparous / viviparous ones " to "ovoviviparity / viviparity" (line 94)

- clarify the meaning of the statement "but the females produced wax" (line 227). Do you mean that, while oocysts were not observed, there was maternally produced wax?

- clarify the meaning of the sentence "so the researchers thought the first nymph was produced by scale insects such as P. solani and another invasive mealybug..." (lines 230-232). Do you mean to say that you wondered if there was some cross contamination from another species? How was this ruled out?

- clarify if "Vennila et al considered the scale insect to be able to reproduce in both ways" (line 233) refers to the same species, P. solani, as this study, or does this refer to a different species?

- change "eggs are under stress" to "eggs under stress" (line 259)

- clarify why the research on yeast-like symbiotes is relevant to the study in question (line 264). Are the authors suggesting that the presence / absence of symbiotes might explain the presence of non-developing eggs?

- clarify the meaning of "no longer resemble an egg on a microscopic level" (line 299). Do they still have an eggshell? Do you mean that the embryo within the chorion is fully developed and close to hatching?

- use more specific language to describe the question at line 319: "Although the reproductive mode of P. solani has been previously described, the actual process may not be simple" (lines 318-319). What remains unknown, prior to the current study?

- the language in the conclusion needs to be reworked to resolve ambiguity. Specifically the phrases: "one thing needs to be made clear" (line 325), "ovoviviparity is defined as the mother giving live birth, basically to larvae" (line 328, and this sentence requires a reference), and "we conclude that the reproductive 'product' of P. solani is an egg" (line 329). As currently written, it is not easy to understand what feature of P. solani reproductive biology makes this species distinct from other ovoviviparous species, and according to which definition of ovoviviparity this claim is being made.

- The legend of figure 1 contains several sentence fragments, which should be corrected, e.g. change "finished observation device, and contained two separate adult females" to "The final observation device contained two separate adult female".

- change "The first 5 minutes" to "After the first 5 minutes", and clarify the meaning of "the hatching process was basically completed" (figure 3 legend).

- typo at "analyze d" (figure 4 legend)

- change "and the vitelline membrance had been detached" to "at the point at which the vitelline membrane detached"; "Eggs hatch under the mother's body" to "Eggs raised under the mother's body"; and change "(F)" to "(E)". (figure 5). Additionally, is that the vitelline membrane or the chorion? Can the authors please clarify how this structure was defined / described?

Experimental design

The manuscript addresses the question of whether the solanum mealybug, like other scale insects, demonstrates ovoviviparity. In addition, the authors describe and investigate a phenomenon of two egg types, one developing toward hatching and one not.

The authors investigate the first question through classic observational description. The descriptions and experimental manipulations they provide are sufficient to answer the question.

The second question is addressed using a variety of techniques, including electron microscopy and RNAseq.

The exact purpose and findings of the electron microscopy and RNAseq experiments should be described in further detail. From my understanding of their findings, the mealybug lays two kinds of eggs: one that develops toward hatching, and one that does not. The authors show that these can be unambiguously identified by the presence of eyespots at the moment of oviposition.

The authors mention a few factors that may explain the presence of these non-developing eggs, including that development may have arrested around the time of blastokinesis or that these eggs never underwent any form of development, and may instead be a nutritive propagule for other offspring to consume. However, their description of the results of the electron microscopy and RNAseq experiments do not help to distinguish between these scenarios.

The authors should seek to answer the following questions using the data they already have:

- From the microscopy data for eggs without-ES, is there any evidence that any development had taken place? Are there any remaining embryonic structures? Could the authors perhaps identify an embryonic stage at which development arrested for the eggs without-ES?

- From the RNAseq data, what genes are active in the eggs without-ES? Do these provide any suggestion of the function of these eggs as nutritive propagules, or as embryos arrested at some stage of development?

- Many insects lay non-viable eggs. Do the authors have any evidence that the scale insect is laying a larger proportion of non-viable eggs than other scale insects? than other parthenogenic species? Is there anything to suggest that these eggs without-ES are a unique structure beyond a non-viable egg?

The authors have gathered a very rich set of data with the electron microscopy and RNAseq experiments, but without more context, these data are essentially not utilized in this manuscript.

Validity of the findings

The authors state that the mealybug does not qualify as ovoviviparous, and suggest an alternative term to describe their findings: semi-ovoviviparous. However their results are not consistent with this finding. I believe this to be due to a misunderstanding of the definitions of these reproductive categories. From the entomological literature, the definition is that "ovoviviparous species oviposit eggs at an advanced stage of embryological development, and the larva exits the egg shell during or immediately following oviposition" (Meier et al 1999, Ovoviviparity and viviparity in the Diptera, Biol. Rev.). This is precisely what the authors describe here, as they state at lines 228-229: "After egg laying, the yolk membrane was detached from the egg (eggshell), followed by the appearance of 1st instar nymphs quickly crawling away from the lower part of the mother. It took a short time (< 30 minutes) to complete this process...". The creation of the category 'semi-ovoviviparity' is therefore unnecessary and would further confuse the situation in describing reproductive categories.

This subject has been discussed thoroughly in the literature (Blackburn 1994, Meier 1999, Weber 1953 vs. Sellier 1955), and my interpretation of this discussion is that there is a continuum between full viviparity (live birth) and full oviparity (eggs laid immediately following fertilization), and that ovoviviparity has been used to variably describe any number of states in between those extremes and along that continuum. Given this, it is most important to describe the actual details of the timing of oviposition, hatching, and development, which the authors have already done nicely. Beyond that, defining the category is important in linking these new descriptions to similar descriptions from the literature, which is where this manuscript currently falls short.

In order to rectify this, I believe that the authors should restate their conclusions to state that they find this species to exhibit ovoviviparity, consistent with other scale insects, or else provide additional and much stronger evidence that this species indeed exhibits a reproductive mode distinct from any ovoviviparous mode previously described. This would require changing the "Discussion" section of the abstract, as well as most of the text of the conclusion.

Additional comments

I appreciated reading this article, and especially enjoyed the excellent use of classical description methods to answer a straightforward question: does P. solani give live birth? I also appreciated the transparency with which the data was shared.

Reviewer 3 ·

Basic reporting

The manuscript of Huang et al. provides new information about the reproductive biology of one of the scale insect species. This information is rather simple and rather usual for parthenogenetic species. All the novelty can be expressed in the phrase “Females lay two sorts of eggs, viable and non-viable, but we do not know why is it so”. Such kind of reports are usually published as “Short communications” and I would advise to reduce the present 23-pages to several pages only, deleting all common information, which the authors abundantly compiled from the well-known literature.

Experimental design

To my mind, methods, which the authors used for study of the material, are not suitable. Main morphological structures (like legs, antennae or eyespots) of the developing embryo of scale insects are clearly visible in the drop of acetic acid or ethanol under the usual stereoscopic microscope; it is not need to use complicated electron microscopy for this purpose. As for the RNA-sequencing I am absolutely unable to understand for what it was need in this case. It is clear without any studies that developing embryos and non-viable eggs have different biochemical composition, but the authors do not really study of this composition and only say that there is the difference in the data of RNA-sequencing. It gives nothing for the understanding of the reason, why two sorts of eggs are laid. The only way to clarify such reason is detail cytogenetic studies of the parthenogenesis, in particular female meiosis in Phenacoccus solani. However, the authors totally avoid any cytogenetic investigations in their work.

Validity of the findings

no comment

Additional comments

In addition, I would advise to be more careful with a terminology. For example, “oocyct” is a resistant, thick-walled spore containing the zygote in Protozoa (see, for example, https://www.biologyonline.com/dictionary/oocyst or any other dictuonary). I am even unable to imagine what the authors mean, applying this term to the scale-insects. Also, the term “semi-ovoviviparity” seems to be rather strange and unclear. Previously the term “incomplete ovoviviparity” was used in the literature, cited by the authors, for the situations when the only part of embryogenesis occurs inside of the mother’s body….

---

## Round 0.2 · Minor Revisions

Dear Dr. Huang and colleagues:

Thanks for revising your manuscript. The reviewers are very satisfied with your revision (as am I). Great! However, there are a few minor edits to make. Please address these ASAP so we may move towards acceptance of your work.

Best,

-joe

Reviewer 1 ·

Basic reporting

I am satisfied with all corrections incorporated into the manuscript. I will be be happy to see the final published version of the paper!

Experimental design

n/a

Validity of the findings

n/a

Additional comments

I strongly suggest acceptance of the paper.

·

Basic reporting

The manuscript is considerably improved, especially in the areas of defining the reproductive mode in relation to past literature, and in providing context for the RNAseq results.

The typo "analyze d" is still present in two places in the legend of Figure 5.

Experimental design

No comment.

Validity of the findings

The statement "These eggs did not hatch, just as a chick cannot hatch without a hen" seems to be in contradiction with the finding "Therefore, we suggest that the mother’s body has no effect on the success of egg hatching". Their results show that the reason that eggs without eyespots did not hatch is not related to the presence / absence of the mother. I suggest that the former statement, "just as a chick cannot hatch without a hen" should be removed.

Additional comments

In my original review I stated "This subject has been discussed thoroughly in the literature (Blackburn 1994, Meier 1999, Weber 1953 vs. Sellier 1955), and my interpretation of this discussion is that there is a continuum between full viviparity (live birth) and full oviparity (eggs laid immediately following fertilization), and that ovoviviparity has been used to variably describe any number of states in between those extremes and along that continuum."

The authors have used the second part of my statement, starting at "there is a continuum...", in the abstract and conclusion of their paper. As stated above, this was intended as my interpretation of the situation, and was not meant to be reproduced as the authoritative word on ovoviviparity. I ask that the authors replace these sentences (lines 49 - 52, and 362 - 365) defining ovoviviparity with one based on the published literature, for example, the publications listed above

Especially, I do not view this statement as a finding from the paper, and so I do not think it is appropriate for the authors to claim "We determine that there is a continuum..." (line 362). This sentence should be removed.

Reviewer 3 ·

Basic reporting

I have no additional comments in comparison with my previous review. The authors provided a row of changes and additions in the text and I think that the paper may be published in the present form. I can not agree with some approaches and some terms, but the authors have a right to express their own views, of course.

Experimental design

no comments

Validity of the findings

no comments

Additional comments

no comments

---

## Round 0.3 · accepted · Accept

Dear Dr. Huang and colleagues:

Thanks for again revising your manuscript. I now believe that your manuscript is suitable for publication. Congratulations! I look forward to seeing this work in print, and I anticipate it being an important resource for research studying reproductive biology of the solanum mealybug. Thanks again for choosing PeerJ to publish such important work.

Best,

-joe